# Traumatic Fracture Treatment: Calcium Phosphate Bone Substitute Case–Control Study in Humerus, Radius, Tibia Fractures—Assessing Efficacy and Recovery Outcomes

**DOI:** 10.3390/biomedicines11102862

**Published:** 2023-10-22

**Authors:** Gero Knapp, Jonas Pawelke, Christian Heiss, Sera Elmas, Vithusha Vinayahalingam, Thaqif ElKhassawna

**Affiliations:** 1Department of Trauma, Hand and Reconstructive Surgery, Faculty of Medicine, Justus Liebig University of Giessen, 35392 Giessen, Germany; christian.heiss@chiru.med.uni-giessen.de (C.H.); sera.elmas@gmail.com (S.E.); 2Experimental Trauma Surgery, Faculty of Medicine, Justus Liebig University of Giessen, 35392 Giessen, Germany; jonas.pawelke@med.uni-giessen.de (J.P.); vithusha.vinayahalingam@med.uni-giessen.de (V.V.); thaqif.elkhassawna@chiru.med.uni-giessen.de (T.E.)

**Keywords:** bone substitute, calcium phosphate, trauma surgery, geriatrics, synthetic bone material, alloplastic, distal radius, proximal humerus, proximal tibia

## Abstract

To date, insufficient investigation has been carried out on the biocompatibility of synthetic bioactive bone substitute materials after traumatically induced bone fractures in clinical conditions. This study encompasses the safety, resorption, healing process, and complications of surgical treatment. Our current hypothesis posits that calcium phosphate-based bone substitutes could improve bone healing. In this retrospective case–control study, over 290 patients who underwent surgical treatment for acute fractures were examined. Bone defects were augmented with calcium phosphate-based bone substitute material (CP) in comparison to with empty defect treatment (ED) between 2011 and 2018. A novel scoring system for fracture healing was introduced to assess bone healing in up to six radiological follow-up examinations. Furthermore, demographic data, concomitant diseases, and complications were subjected to analysis. Data analysis disclosed significantly fewer postoperative complications in the CP group relative to the ED group (*p* < 0.001). The CP group revealed decreased risks of experiencing complications (*p* < 0.001), arthrosis (*p* = 0.01), and neurological diseases (*p* < 0.001). The fracture edge, the fracture gap, and the articular surface were definably enhanced. Osteosynthesis and general bone density demonstrated similarity (*p* > 0.05). Subgroup analysis focusing on patients aged 64 years and older revealed a diminished complication incidence within the CP group (*p* = 0.025). Notably, the application of CP bone substitute materials showed discernible benefits in geriatric patients, evident by decreased rates of pseudarthrosis (*p* = 0.059). Intermediate follow-up evaluations disclosed marked enhancements in fracture gap, edge, and articular surface conditions through the utilization of CP-based substitutes (*p* < 0.05). In conclusion, calcium phosphate-based bone substitute materials assert their clinical integrity by demonstrating safety in clinical applications. They substantiate an accelerated early osseous healing trajectory while concurrently decreasing the severity of complications within the bone substitute cohort. In vivo advantages were demonstrated for CP bone graft substitutes.

## 1. Introduction

In the realm of orthopedic and traumatological surgery, addressing bone defects poses a significant challenge in surgical management.

Autologous bone grafting remains the gold standard for augmenting osseus defects. However, contemporary discussions center on the risks associated with a second surgical intervention for graft harvesting. Common harvesting sites, such as the iliac crest or proximal tibia, carry additional risks for the donor site of the patient [1,2]. Palmer et al. reported a complication rate of 73.3% for iliac crest bone substance harvesting [3], with no significant disparities in complication incidence across evaluated regions [4]. The literature suggests that the use of autologous bone substitute materials may lead to reduced infection in rats and improved bone healing [5].

Alloplastic bone grafts serve as a viable alternative for defect augmentation, offering advantages such as unlimited availability and the absence of a need for a second surgical site [6]. In comparison to allogenic bone substitutes, synthetic bone material carries no risk of disease transmission and is less antigenic [7]. However, alloplastic bone graft substitutes are associated with higher monetary costs when compared to autologous materials [8,9]. In contrast, they offer reduced physical and monetary costs in terms of defect filling [8,10]. Regarding the bioactivity of bone substitutes, the comparative bio-resorption of autologous bone substitutes is under critical examination [11]. Some researchers propose improved healing processes and bridging using alloplastic bone substitutes, while others report worse healing processes and increased complications due to the absence of biocompatibility [12]. Alloplastic bone substitute materials provide biological stability and volume retention for cellular infiltration [13], resulting in clinical advantages such as reduced comorbidity and an expedited postoperative healing phase [14]. Notably, Millward et al. reported a reduced risk of infection-associated complications in cranioplasty implants [15], whereas Nisyrios et al. reported an increased risk of infection in in vitro examinations [16].

In summary, the data landscape for alloplastic bone graft substitutes remains inconsistent, with a notable lack of clinical studies. Although in vitro and animal studies are available for most materials, the authors of previous studies recommend further examinations, given the differences in bone metabolism in animals. Under the new Medical Device Regulation (MDR) implemented in 2022, clinical studies are now a requirement for the approval of bone graft substitutes in Europe [17].

Alloplastic bone substitutes can be composed of various materials, primarily different mono- or poly-forms of minerals used to create substances resembling bone. Key components such as calcium sulfate (CS), calcium phosphate (CP), mixtures of calcium sulfate and calcium phosphate (CSCP), demineralized bone matrix (BM), or composites of tricalcium phosphate with (nanocrystalline) hydroxyapatite (NHA) are used [18].

This study investigates Calcibon^®^ (Zimmer Biomet Deutschland GmbH, Freiburg, Germany), an injectable, resorbable bone graft substitute with a composition similar to normal bone substance. Surgeons receive a viscous paste, composed of a powder mixed with a liquid emulsion. The composition includes phosphates, hydrous disodium hydrogen phosphate (NaHPO_4_), and the specific powder. After blending both components, the optimal liquid to powder ratio was determined to be 0.35 mL of liquid per gram. The liquid emulsion initiates the bone substitute curing process. To ensure that the structure of Calcibon^®^ emulates that of biological bone tissue, diverse materials are employed, including alpha-tricalcium phosphate (Alpha-TCP), calcium hydrogen phosphate (CaHPO_4_), calcium carbonate (CaCO_3_), and hydroxyapatite (HA). Mixing these components initiates an endothermic reaction [17].

The microstructural properties of Calcibon^®^ play a pivotal role in influencing its mechanical strength, facilitating vessel ingrowth, and promoting cell growth. Against this backdrop, the microstructure and biomechanical properties of Calcibon^®^ were considered. Notably, micro-CT analysis revealed specific characteristics, including a typical pore size of 41.6 µm, with a maximum size measuring 230 µm. Significantly, more than 95% of the pores were smaller than 125 µm. This substantial open porosity, representing the proportion of open pores relative to the total volume of the bone graft substitute, was quantified at 0.22 ± 0.75%. Additionally, the stability of Calcibon^®^ was assessed by calculating the modulus of elasticity, yielding values of 790 ± 132 MPa (Table 1) [19].

Detached bone cells present a common challenge when assessing the effectiveness of Calcibon^®^ in experimental settings. Over a 12-day cell culture period, researchers observed detached cells and subsequent cell death, attributed to reduced DNA content in the surrounding cells. This phenomenon is linked to diminished populations of viable cells within the acidic environment created by the release of calcium and phosphate ions from the TCP in the Calcibon^®^ matrix under laboratory conditions [17]. Clinical studies involving Calcibon^®^ have been limited, with the majority focusing on degenerative diseases. Insufficient research has explored the application of this bone graft substitute in traumatic fractures of the thoracolumbar junction [25,27,28,29]. Notably, patients enrolled in clinical trials reported pain relief (evaluated using the Visual Analog Scale) and an early return to normal activities. In contrast to in vivo and in vitro studies, clinical observations indicated less resorption of the bone graft substitute. However, significant complications were documented in each study [25,28,29], including a loss of reduction, infection, and increased pain.

Material testing of Calcibon^®^ did not reveal any toxic effects. Histological examination indicated no infections and demonstrated highly favorable stimulations of mesenchymal stem cell growth compared to other calcium phosphate-based bone substitute materials [25].

In summary, while Calcibon^®^ has undergone extensive investigation concerning its microstructural and biomechanical properties, there remains a remarkable lack of research in the realm of clinical utilization, especially regarding trauma-related bone defects.

This study contributes to our clinical understanding of Calcibon^®^ by comparing the treatment of empty defects with bone substitute implantation. The authors hypothesize that the bone substitute material enhances the bone healing process by supporting biological bone healing.

## 2. Materials and Methods

This study encompassed a cohort of 295 patients who underwent surgical intervention for traumatic fractures at a level 1 trauma center during the period spanning 2011 to 2018. The study included cases of acute fractures involving the proximal humerus, distal radius, and proximal tibia. All enrolled patients received surgical treatment in accordance with the established regional gold standard for fracture management Exclusion criteria encompassed patients with pathological fractures, and those requiring surgical defect vault filling due to bone tumors were excluded. Additionally, patients with contraindications for anesthesia or surgery (such as pregnancy, life-threatening diseases, or infections) and fractures in children were excluded from the study. Consequently, the study focused on patients aged 17 years and above, with either a closed or arrested physis. After implementing these exclusion criteria, the final evaluation cohort consisted of 290 patients (Figure 1).

All patients included in this study were categorized into two distinct cohorts. The interventional group underwent osteosynthesis in addition to defect vault augmentation, utilizing calcium phosphate-based bone substitute materials (CP) and Calcibon^®^ (Zimmer Biomet Deutschland GmbH, Freiburg, Germany). In contrast, the alternate group underwent surgical intervention without supplementary bone substitute materials, characterized as the empty defect treatment (ED).

A total of 136 patients were enrolled in the CP group, while 154 patients in the ED group were selected based on gender, age, and fracture morphology during the same study period spanning from 2011 to 2018. Comprehensive evaluations of patient files were conducted to assess medical history and comorbidities.

Pre-existing conditions were assessed using the American Society of Anesthesiologists Physical Status System (ASA classification) [30]. Furthermore, the Carlson Comorbidity Index (CCI), a contemporary classification system for comorbidities, was employed to ensure comparability between the ED and CP groups.

In terms of accident mechanism, injuries were stratified into high-impact and low-impact injuries. High-impact injuries included falls from a height, traffic accidents, and sports injuries, each characterized by their corresponding kinematics. Low-impact trauma encompassed stumbling, slips, and falls (SSF) accidents, domestic falls, and falls on stairs (less than 5 steps). Demographic data including sex, age, weight, body mass index, type of accident, region, and side of the fracture were obtained from the patients’ medical records.

Fracture severity was reassessed using the preoperative radiographs, initially considering the number of open fractures.

Open fractures were classified based on the Tscherne and Oestern criteria [31], taking into account the extent of skin tissue tear. Additionally, the Association of Osteosynthesis (AO) classification was used for fracture classification. For long bone fractures, a numeric code was assigned (humerus (1), radius (2), and tibia (3)). Fractures were further categorized as intraarticular (C), partial intraarticular (B), or extraarticular (A). A numeric code was used to determine the fracture morphology.

Postoperatively, up to five radiographs or CT scans were reviewed per patient. In clinical practice, physical examinations were complemented by imaging to assess fracture healing. Drawing from previous studies on bone healing in both acute and chronic defects (Bohndorf et al., 2006, Freyschmidt et al., 2016, and Islam et al., 2000), a novel classification system was developed for the evaluation of bone healing on radiographs and CT scans (Figure 2) [32,33,34,35,36]. This classification system aimed to assess the fracture healing process on an ordinal scale according to the German school grading system (1–6, with 1 being the best and 6 the worst).

The fractures were divided into four areas for evaluation (Appendix A). Initially, the fracture edge was sharply defined in the initial days post-trauma (grade 5). Subsequently, due to fracture displacement and the bone remodeling process, the bone density within the fracture gap diminished, resulting in a visible, pronounced fracture gap (grade 5). Following the acute phase, the fracture edge lost its sharpness within three to four weeks. Radiographs displayed the loss of sharp edges (grade 4) and indistinct bone gaps (grade 4). Meanwhile, compaction within the fracture gap and faintly visible fracture edges indicated a state preceding optimal healing (grade 2). If optimal osseous fracture healing (restitutio ad integrum) was achieved, the highest grade (grade 1) was assigned. It is noteworthy that four to six months are typically required for optimal bone healing [34]. Variations in the articular surface and differences in the osteosynthesis material were also considered (Appendix A).

In addition to the previously described grading system, we conducted assessments of bone substance and bone density to predict the risk of periosteosynthetic fractures and evaluate the activity status. Moreover, we meticulously evaluated significant pathologies associated with the use of bone substitutes, including expanding fracture edges, increased fracture gaps, and the emergence of new fracture lines. To determine the presence of arthrosis, we considered factors such as subchondral sclerosis, osteophytes, and the minimization of the articular gap, relying on subjective evaluations.

In our daily clinical practice, we diligently documented complications that could impact treatment outcomes. These complications included pseudarthrosis and infections, each with distinct definitions. According to the FDA’s definition, pseudarthrosis requires nonunion to persist for more than six months with no change in the healing process over three months [37]. In the classification of infections, we differentiated between deep infections involving the implanted bone substitute, muscle, bone substitute, and osteosynthesis, and superficial infections limited to the skin. Superficial dermal wound infections were considered independent of the implant. For comprehensive coverage, we also recorded neurological complications such as persistent postoperative pain, hyperesthesia, dysesthesia, and paresthesia (lasting more than six weeks). Additionally, when assessing long-term complications, we observed post-traumatic cartilage damage (including post-traumatic arthrosis, osteoarthritis, and chondromalacia), with clinical and radiological criteria taken into account. Radiological criteria were particularly emphasized in the evaluation of pseudarthrosis and infection.

For the purpose of statistical analysis, radiological variables were assigned ordinal scales, while demographic data and complications were categorized nominally. Subsequent statistical analyses were conducted using IBM^®^ SPSS^®^ Statistics (IBM SPSS Statistics for Apple, Version 28.0. Armonk, NY: IBM Corp). To compare and determine significance between the ED and CP groups, we primarily employed the Mann–Whitney test, and, in some cases, Kolmogorov analysis was applied. In the case of equal distribution, the t-test for independent samples was utilized. The distinction between equal and unequal distribution was assessed based on kurtosis near 0 and skewness near 3 in cases of equal distribution. The significance level was set at *p* ≤ 0.05. When feasible, the level of significance was further restricted to *p* ≤ 0.005 to enhance the analysis and reduce the number needed to treat (NNT).

To evaluate the statistical power post hoc for two independent groups, which were examined using the Wilcoxon–Mann–Whitney test, we analyzed a sample size of 137 patients in the calcium phosphate-based bone augmentation group (Calcibon^®^) and 153 patients in the empty defect group. This analysis demonstrated a robust power value of 0.99, employing the G*Power (G*Power, Version 3.1, Erdfelder, Faul, and Buchner, HHU Düsseldorf, Germany) program.

## 3. Results

A total of 295 patients with acute fractures were treated at the Department of Trauma Surgery of the University Hospital Giessen and Marburg, between 2010 and 2017. Out of these, 290 patients were included due to the specification of acute fractures in adults and at three locations within long bones: the proximal humerus, distal radius, and proximal tibia.

### 3.1. Demographic and Descriptive Parameters

In this single-center trial, the evaluation encompassed 156 females (53.8%, n = 156) and 134 males (46.2%, n = 134). There were no differences between female and male patients in both treatment groups (*p* = 0.81). The average age was 55.44 years, with no statistically significant difference between the two groups (Minimum: Maximum; Mean ± SD, 17:91; 55.44 ± 17.69). Patients in the ED group (17:91; 53.24 ± 18.64) were insignificantly younger than those in the CP group (18:89; 57.90 ± 16.28) (*p* = 0.06).

For the classification of pre-existing comorbidities using the ASA classification system, patients were distributed across ASA 1 to ASA 4 (n = 288). The mean comorbidity evaluation was ASA 2.06, showing no difference between the two treatment cohorts (*p* = 0.46). To specify comorbidities, the CCI classification system revealed values of 0 to 13, with a mean of 2.78, and no differences were noted between the ED and CP cohorts (*p* = 0.39).

The majority of patients experienced fractures in the upper extremity (60.7%, n = 176). The distal radius was the most affected site (87.5%, n = 154), followed by proximal humeral fractures (12.5%, n = 22). Fractures in the lower extremity were observed in 39.3% of cases (n = 114), exclusively involving the proximal tibia (100%, n = 114). Specifically, within the ED group, 58.8% (n = 90) had distal radius fractures, 7.2% (n = 11) had proximal humeral fractures, and 34.0% (n = 52) had proximal tibial fractures. In the CP group, 46.7% (n = 64) had distal radius fractures, 8.0% (n = 11) had proximal humeral fractures, and 45.3% (n = 62) had proximal tibial fractures. No significant difference was observed between the two cohorts (*p* = 0.06).

Of the injuries, 175 (60.3%) occurred on the left side, while 114 (39.3%) fractures were sustained on the right side of the body. The analyzed body mass index (BMI) showed a typical European mean (17.10:44.08; 26.89 ± 4.90). Subjects in the ED group had a BMI of (18.14:44.08; 27.15 ± 4.99), while those in the CP group had (17.10:41.60; 26.60 ± 4.81), with no significant differences compared to the ED cohort (*p* = 0.20).

Descriptive parameters were recorded without significant differences between both treatment cohorts (*p* > 0.05).

The duration of hospital stay was (0:72; 10.66 ± 11.51). A significant difference was observed in the duration of hospital stay (*p* = 0.007), with shorter stays for patients treated with CP augmentation (1:72; 10.34 ± 10.81) compared to patients treated in the ED group (0:70; 10.94 ± 12.13).

Fracture severity and morphology were classified based on the AO classification. In both groups, 12.8% (n = 37) had extraarticular, 20.0% (n = 58) had partial intraarticular, and 64.8% (n = 188) had intraarticular fractures. Meanwhile, 2.4% (n = 7) fractures could not be examined due to missing preoperative radiographs. In the ED group, 15.7% (n = 24) had extraarticular, 12.4% (n = 19) had partial intraarticular, and 70.6% (n = 108) had intraarticular fractures. In the CP cohort, the distribution of fractures based on severity and morphology was as follows: 9.5% (n = 13) were extraarticular, 28.5% (n = 39) were partial intraarticular, and 58.4% (n = 80) were intraarticular fractures (Figure 3). When comparing the severity of fractures between the ED and CP treatment cohorts and calculating significance, no significant differences were found (*p* = 0.19).

A significant difference was observed in the incidence of collateral fractures in the forearm (ulna fractures) and the lower leg (fibula fractures) between the ED group and the group treated with CP augmentation (*p* = 0.006). A higher prevalence of collateral fractures was noted in the group without augmentation (63.4%, n = 97), whereas, in the CP group, the incidence was lower (46.0%, n = 63).

The causes of the fractures were divided into low-impact trauma (45.5%, n = 132) and high-impact trauma (49.7%, n = 144). Anamnesis was missing in 4.8% (n = 14) of cases.

Within the low-impact accident subgroup, 27.2% (n = 79) were attributed to stumbling, 12.8% (n = 37) to domestic accidents, and 5.5% (n = 16) to falls on stairs. High-impact fractures resulted from traffic accidents (23.1%, n = 67), including car accidents (12.4%, n = 36), sports injuries (13.8%, n = 40), and high-impact falls (12.8%, n = 37). Nevertheless, 4.8% (n = 14) fractures could not be categorized due to missing anamnesis. In the ED group, high-impact trauma accounted for the majority of fractures (53.6%, n = 82), while low-impact trauma caused 41.2% (n = 63) of cases. Moreover, 5.2% (n = 8) of fractures had an indeterminate impact due to missing anamnesis. In the CP augmentation group, nearly half of the fractures were due to low-impact accidents (50.4%, n = 69), with high-impact injuries at 45.3% (n = 62) of fractures. Meanwhile, 4.4% (n = 6) of the patients’ files lacked trauma categorization.

Utilizing a case–control study to assess the impact of adding Calcibon^®^ augmentation in surgery for bony defect filling compared to the empty defect group, 47.2% (n = 137) of patients received additional CP augmentation with a mean injected volume of (1.0:7.5; 2.26 ± 1.45) mL, while 52.8% (n = 153) of patients underwent treatment without added bone substitutes.

### 3.2. Clinical Outcome—Complications

In terms of clinical outcomes and complications, we observed various postoperative complexities, with a range and mean of (0:4; 0.5 ± 0.74) complications per patient. Specifically, fractures treated without an added bone graft showed significantly more postoperative complications (0:3; 0.64 ± 0.74) compared with the CP group (0:4; 0.34 ± 0.70) (*p* < 0.001). These complications included pseudarthrosis and other orthopedic issues. Notably, significant differences were observed in comparing (1) pseudarthrosis, (2) other complications, and (3) no complications between the CP augmentation and ED treatment groups. Overall, the use of the CP bone substitute showed an improvement in complications (*p* < 0.001). Significance was also evident between the ED and CP groups concerning pseudarthrosis and no pseudarthrosis (*p* = 0.022) (Figure 4). Additionally, there were fewer instances of post-traumatic arthrosis (*p* = 0.01) and neurological issues (*p* < 0.001) in the CP augmentation group, as indicated in the overview of complications (Figure 5).

Many other complications did not show a significant difference between the two groups, including necrosis and infection, ligamentous and muscular damage, osteoporosis due to inactivity, psychological disorders, delayed bone healing, and premature removal of osteosynthesis (*p* > 0.05). Importantly, no patient experienced premature death due to augmentation (Table 2).

### 3.3. Radiological Outcome

When measuring the bone healing process through postoperative radiographs, an improved healing process was demonstrated when using a CP bone graft substitute. Up to six radiographs taken on various postoperative days were evaluated. The time points for the radiographs were as follows: the first period (n = 282) was at (0:7; 1.84 ± 1.18) days, the second time (n = 254) at day (10:79; 27.06 ± 11.19), the third radiograph (n = 233) at day (40:131; 61.99 ± 20.83), followed by (n = 171) at (126:193; 106.84 ± 34.94) days, (n = 126) at (126:571; 285.26 ± 76.40) days, and finally (n = 98) after (234:1986; 645.02 ± 326.48) days. A difference was observed when comparing the fracture edge in the ED group with the CP augmentation, except for the first and the last follow-up (FU) examinations (FU1 *p* = 0.53; FU2–5 *p* < 0.001; FU6 *p* = 0.126), with decreased grades in the CP group. The same pattern was detected when considering the gap in the fracture. When using CP augmentation, there was a significant increase in the density level within the fracture gap (FU1 *p* = 0.04; FU2–5 *p* < 0.001; FU6 *p* = 0.02) in all radiographs. Furthermore, the articular surface in the CP augmentation group showed improved grading compared to the ED group (FU1–4 *p* < 0.001; FU5 *p* = 0.002; FU6 *p* = 0.001). Insignificant differences were observed between the ED and CP groups when assessing osteosynthesis and the bone substance (*p* > 0.05). The rate of revision surgery, including subsequent surgical procedures and metal removal, did not show a statistically significant difference between the two groups (*p* = 0.96). Additionally, no statistical significance (*p* = 0.53) was found between the groups regarding osteosynthesis removal (Table 3).

However, it is important to note that a potential confounder in this study is the fact that most patients were geriatric patients, who are known to experience decreased bone healing. Additionally, collateral bone fractures were observed in the CP group, which can lead to an increased length of hospital stay.

### 3.4. Subgroup Analysis—Geriatric Patients

Focusing on geriatric patients over 64 years old, an analysis was conducted due to the assumption of an increased incidence of osteoporosis and higher infection rates in this demographic. This subgroup consisted of 97 patients, with 50.5% (n = 49) of patients treated in the ED group and 49.5% (n = 48) in the CP augmentation group. Among these patients, 85.6% (n = 83) were female and 14.4% (n = 14) were male. Demographic data showed similarities between the ED and CP groups (*p* > 0.05). The majority of patients (61.9%, n = 60) did not experience complications. Meanwhile, 29.9% (n = 29) of patients had one complication, 7.2% (n = 7) of patients had two, and 1.0% (n = 1) of patients had four complications, with fewer complications observed in the CP treatment group (*p* = 0.025). Pseudarthrosis was less prevalent in the CP group (2.1%, n = 1) compared to the ED group (14.3%, n = 7) (*p* = 0.059). There was no significant difference in infection rates between the groups (*p* = 0.62). An improved bone healing process was observed when comparing the fracture edge. Intermediate follow-up examinations showed decreased grades (FU1–2 *p* > 0.05; FU3–5 *p* < 0.05; FU6 *p* = 0.29). The bridging of the fracture gap, as measured by evaluation, significantly improved (FU1 *p* = 0.074; FU2–6 *p* < 0.05). The healing process of the articular surface in the fracture region was better (FU1 *p* = 0.109; FU2–4 *p* < 0.05; FU5–6 *p* > 0.05). A comparison of osteosynthesis and bone density showed no significant differences between the ED treatment and CP augmentation groups (*p* > 0.05).

## 4. Discussion

This study aimed to provide a comprehensive evaluation of the clinical and radiological outcomes associated with the use of an injectable calcium phosphate-based bone substitute material, Calcibon^®^.

### 4.1. Summary of Findings

Upon analyzing the demographic and descriptive parameters of both cohorts of treatment, the empty defect (ED) and calcium phosphate (CP) augmentation groups, we observed no statically significant differences (*p* > 0.05). This favorable comparability was extended to gender, age, the affected body side, and pre-existing conditions assessed using the American Society of Anesthesiologists (ASA) and Charlson Comorbidity Index (CCI) classifications.

Remarkably, the calcium phosphate-based augmentation group exhibited significantly fewer postoperative complications compared to the empty defect treatment group (*p* < 0.001). These complications included a lower incidence of pseudarthrosis (<0.001), reduced post-traumatic arthrosis (*p* = 0.01), and a decreased occurrence of neurological diseases (*p* < 0.001) following surgery. Notably, there were no significant differences in other complications, such as infections and necrosis, between the two treatment options (*p* = 0.71).

Despite the importance of addressing bone tumor defects, they were excluded from our study as our primary focus was on acute fractures and their outcomes, which represent a distinct clinical scenario from bone tumors. The exclusion of bone tumor defects ensured the homogeneity of our study population and allowed for a more focused analysis of fracture-related outcomes.

To achieve lasting stability, addressing the bone healing process is imperative. We demonstrated an improvement in the bone healing process with Calcibon^®^, particularly in the aspects of fracture gap closure, fracture edge stabilization, and articular surface preservation.

### 4.2. Comparison of Complications to Previous Trials

The primary objective of using Calcibon^®^ is to provide long-term resorption and stability, addressing microstructural and macroscopic requirements. One of the key goals when utilizing bone substitute materials is to provide enhanced stability, enabling early mobilization and reducing complication rates [38]. Additionally, a study involving 28 patients showed no loss of reduction even after 10 years [39], supporting the notion of effective fracture zone stabilization. Based on the results, this study supports the assumption of stabilizing the fracture zone. The results of our study support this concept by demonstrating improved stability based on radiological criteria, including the fracture edge and fracture gap. Furthermore, there were no significant differences in postoperative dislocations between the two treatment groups (*p* = 0.80).

In terms of complications, the calcium phosphate-based bone substitute Calcibon^®^ exhibited a notable reduction in their occurrence. Specifically, the incidence of pseudarthrosis and neurological complications (e.g., chronic pain or paresthesia, hypesthesia, or hyperesthesia) was significantly lower in the Calcibon^®^ group (*p* < 0.001). This finding aligns with previous studies that reported improved pain relief and functional outcomes, particularly in fatigue fractures of vertebral bodies, following Calcibon^®^ kyphoplasty [27,40]. Furthermore, bone substitute augmentation has consistently demonstrated improved patient outcomes [29]. Thus, our clinical retrospective study corroborates and supports these prior findings.

While some studies have reported increased infection rates associated with the use of autologous or alloplastic bone grafts [6,41], our study found no significant difference in post-surgery infections between the Calcibon^®^ augmentation group and the empty defect group (*p* = 0.71). In vitro studies have suggested that optimal vascular ingrowth is linked to fewer infections, and the reduced pore size of Calcibon^®^ (41.6 ± 22.0 μm) does not appear to be clinically significant. The ideal pore size for vascular ingrowth in bone substitute materials, according to previous laboratory investigations, is 300 μm or larger [19,42,43]. A pore size of less than 50 μm could be problematic for vascular ingrowth and clinical bone healing [19]. This underscores the importance of ensuring sterile augmentation, which seems more critical than the pore size.

Post-traumatic arthritis has been reported in up to 31% of distal radius fractures within 0–36 months of follow-up, escalating to 64% after 36 months [44]. However, our study identified fewer cases of this, particularly in the Calcibon^®^ group (*p* = 0.01). This suggests that fracture edge stability is crucial in reducing the incidence of post-traumatic arthritis. Contrary to the initial assumptions, improved stability without secondary dislocations and optimal fracture fragment alignment, achieved through bone substitute augmentation, can effectively prevent post-traumatic arthritis.

### 4.3. Comparison of Radiological Bone Healing to Previous Studies

Our investigation revealed that the use of the calcium phosphate-based bone substitute material had a substantial positive impact on the radiological bone healing process. This substantiates our hypothesis that alloplastic bone material substitution enhances the bone recovery process. Specifically, we observed significant improvements in the intermediate radiological outcomes related to fracture edges and fracture gap closure (*p* < 0.05). Most radiographs showed either insignificant or less significant differences during the initial follow-up examination, which was conducted approximately 1.84 days post-surgery. Meanwhile, some prior studies have indicated that the bone healing process was initiated ten days after surgery [34]. We noted that differences between the ED group and the CP group diminished in the follow-up examination over nearly two years (645.02 days).

Supplementing our clinical findings with microstructural and in vitro studies, our research provides compelling evidence for the enhanced bone healing process facilitated by calcium phosphate-based bone graft. Nevertheless, it is essential to acknowledge the retrospective nature of this study, which limits the strength of our conclusions. We recommend future investigations involving prospective randomized controlled clinical trials and matched clinical trials comparing alloplastic bone grafts to the gold standard of autologous bone harvesting from the iliac crest. Furthermore, given that our study was conducted at a single-level trauma center, the external validity could be potentially constrained. A multicenter study would enhance the external validity.

### 4.4. Comparison of Geriatric Patients

Due to the assumption of higher rates of low activity and osteoporosis in geriatric patients, it was imperative to examine the bone healing process within this subgroup [45].

Notably, patients older than 90 years have exhibited pseudarthrosis rates as high as 58.6% [46,47]. In our study, fewer cases of pseudarthrosis were observed in the Calcibon^®^ augmentation group (*p* = 0.059) within the geriatric patient subgroup. This finding suggests that bone substitute augmentation effectively supports the bone healing process in older individuals, even in the face of age-related challenges.

Chronic inflammation, often referred to as inflammaging, is characterized by increased levels of pro-inflammatory markers such as IL-1, TNF, IFN, and IL-6 in geriatric patients. These factors contribute to post-surgery complications, including infections [48]. In our examination of infection rates, which could potentially be exacerbated by foreign materials like Calcibon^®^, we found no significant differences between the treatment cohorts within the geriatric patient subgroup (*p* = 0.62).

### 4.5. Limitations of the Study

Several limitations should be acknowledged due to the retrospective nature of this examination of the calcium phosphate-based bone substitute material, Calcibon^®^. While the examiner was blinded to the patient assignment to the treatment groups, randomization was not feasible, potentially introducing selection bias. However, it is crucial to emphasize that the surgeon was unaware of which patients would later be included in the study. Additionally, conducting this trial at a single-level trauma center may have had implications for the external validity, which could be addressed in a subsequent multicenter study. Furthermore, our evaluation extended from 1.84 days post-surgery up to a two-year follow-up examination. To comprehensively document all complications, future research with a long-term follow-up, extending up to 10 years post-surgery, would yield additional valuable insights. Lastly, it is worth noting that our assessment focused on radiographic examinations to evaluate bone healing and clinical evaluations of complications including pseudarthrosis, infection, and neurological diseases. Incorporating additional clinical parameters such as functional outcomes, patient subjective measures, range of motion, and quality of life in future studies would further enhance the generalizability of our findings.

## 5. Conclusions

The results of this study indicate a significant reduction in complications in the CP group, accompanied by improved radiological bone healing processes.

Alloplastic calcium phosphate-based bone grafts, exemplified by Calcibon^®^, warrant consideration as an alternative to the gold standard of autologous bone graft harvesting from the iliac crest. The findings contribute to our understanding of the benefits and potential applications of calcium phosphate-based bone substitute materials in orthopedic surgery. Future prospective studies and clinical trials are needed to confirm and extend these observations, particularly in different patient populations and clinical contexts.

## Figures and Tables

**Figure 1 biomedicines-11-02862-f001:**
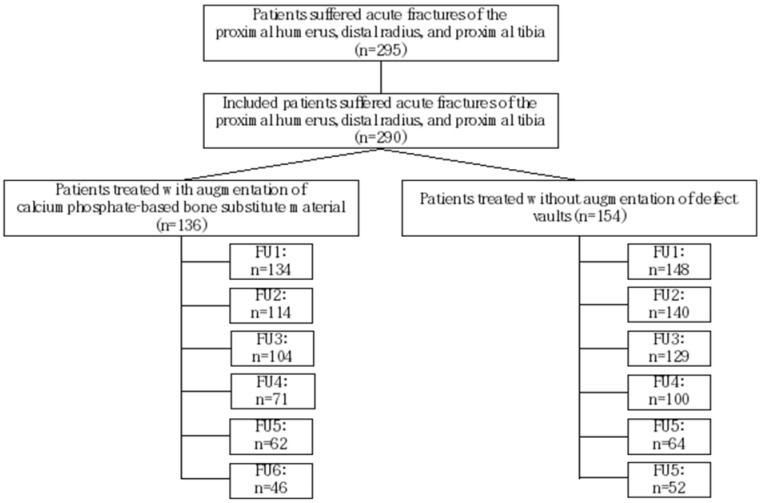
Flow diagram of the participants. Exclusion of 5 patients due to loss of data, missing radiographs, or age younger than 17 years. (Abbreviations: follow-up examination—FU.).

**Figure 2 biomedicines-11-02862-f002:**
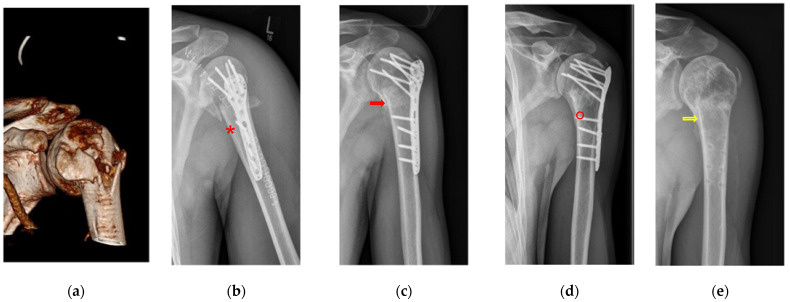
Fracture morphology grading system determined the grade based on the criteria of the bone healing process. (**a**) Morphology of a simple articular fracture prior to surgery (AO classification 11C3.1x). (**b**) Borders of fracture are sharply edged with a wide fracture gap and a loss of density (*). (**c**) Dimly visible edges of the fracture border with a slightly increasing density in the fracture gap (red arrow). (**d**) Advanced bone healing process shows increased bone density in the fracture gap without fracture borders (°). (**e**) Optimal osseus healing process of the fracture borders with increased local bone density in the region of the fracture gap (yellow arrow).

**Figure 3 biomedicines-11-02862-f003:**
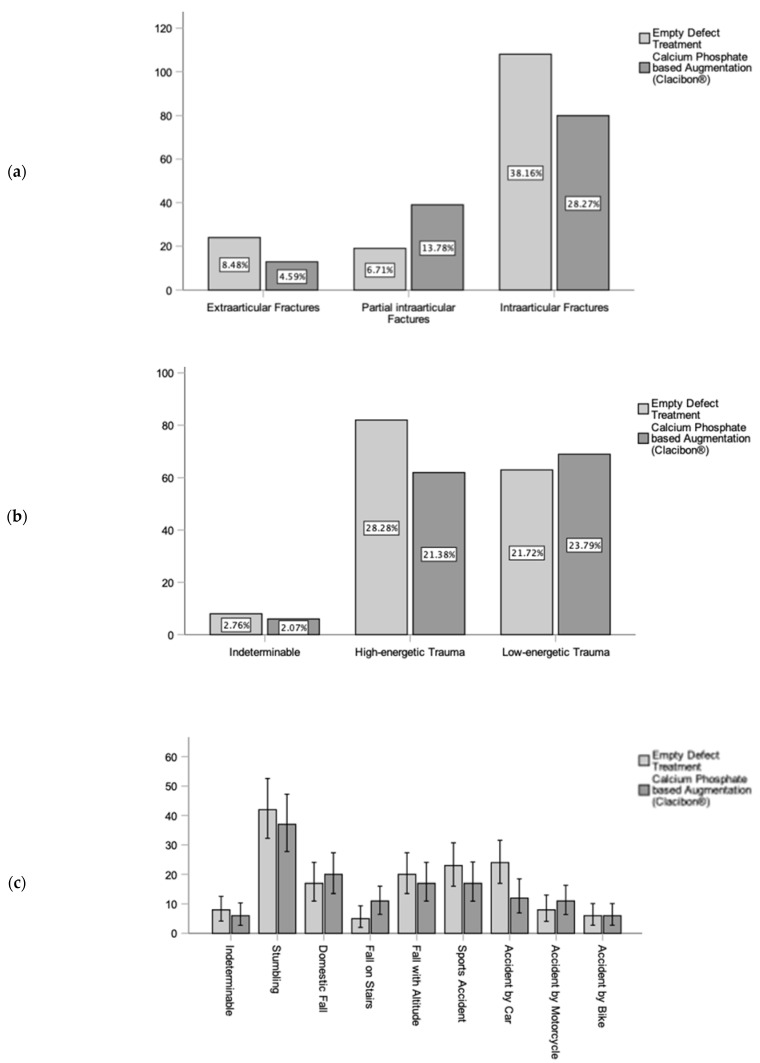
Frequency analysis of location, severity, and cause of injury shows homogeneity of the empty defect (ED) treatment (light grey) and the calcium phosphate augmentation (CP) group (dark grey) (*p* > 0.05). (**a**) Results of fracture location based on the AO classification demonstrate similarity. (**b**) Severity of the accident, divided into low- and high-energetic trauma. (**c**) Cause of fracture demonstrated similarity between both groups of treatment.

**Figure 4 biomedicines-11-02862-f004:**
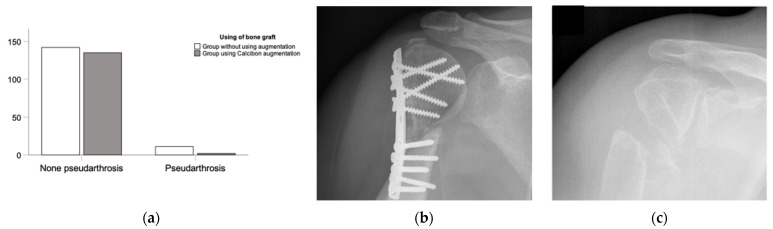
Overview of pseudarthrosis post-surgery. (**a**) Rate of pseudarthrosis in the ED group compared with the rate of pseudarthrosis in the NHA group with significant differences. (**b**) An example of the clinical examination of a recent fracture treatment. (**c**) Conventional radiograph of a pseudarthrosis 18 years post-trauma by X-ray.

**Figure 5 biomedicines-11-02862-f005:**
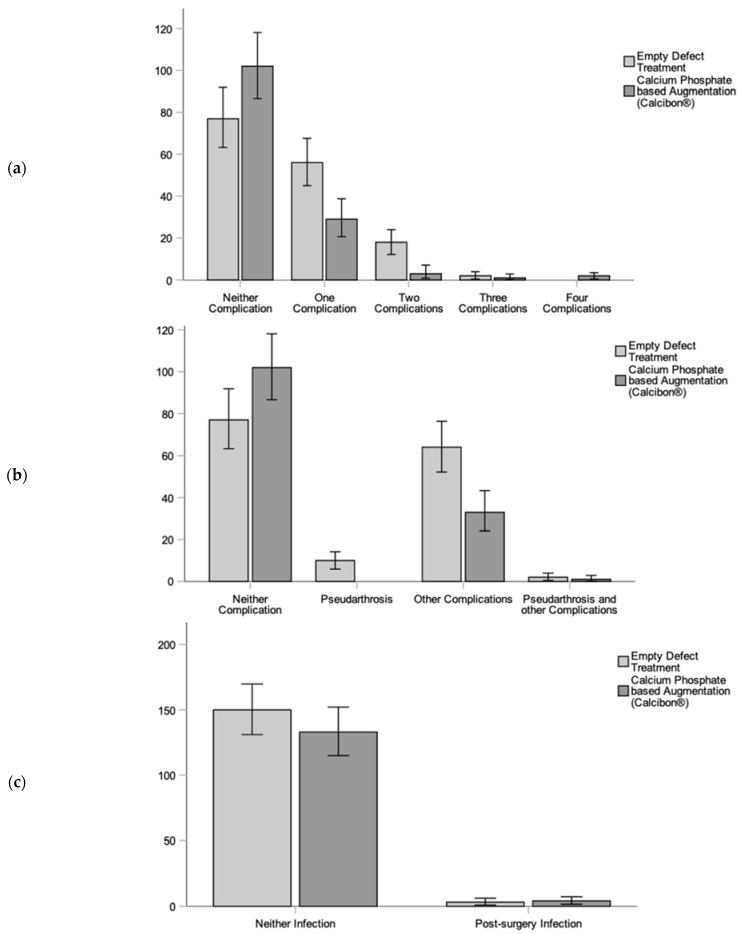
Summary of selected complications demonstrating the differences between the empty defect (ED) treatment (light grey) and the calcium phosphate augmentation (CP) group (dark grey). (**a**) Number of all complications in the ED and the CP cohort with significant differences (*p* = 0.001). (**b**) Comparing pseudarthrosis and other complications in ED and the CP cohort demonstrated less pseudarthrosis and fewer other complications in the CP treatment cohort (*p* < 0.001). (**c**) Infection and necrosis demonstrated insignificantly fewer infections in the ED group compared to the CP treatment group.

**Table 1 biomedicines-11-02862-t001:** Characteristics of calcium phosphate-based bone substitute material, Calcibon^®^, compared with normal bone structure.

Parameter	Calcibon^®^	Adult Bone
Pore size	30–40% pores < 1 μm [17]	One hundred forty-four [20]
	41.6 ± 22.0 μm [19]	
	Pores up to 230 μm [19]	
Open porosity	0.22 ± 0.75% [19]	50–90% trabecular bone
		3–12% cortical bone [21,22]
Density	0.03/cm^3^ [19]	1.85 ± 0.06 g/cm^3^
		0.30 ± 0.10 g/cm^3^ [21,22,23]
Compressive strength	33.95 ± 6.75 MPa [19]	221 MPa [24]
Young’s modulus	790 ± 132 MPa [19]	
Specific surface	100 m^2^/g [25]	
Composition	62.5% α-TCP, 26.8% dicalcium phosphate dihydrate, 8.9% calcium carbonate, 1.8% precipitated hydroxyapatite [17,19]	95% collagen type I [26]

**Table 2 biomedicines-11-02862-t002:** Overview of complications and significance suffered in both groups of treatment. (* in case of significance with *p* < 0.05, ** in case of high significance level with *p* < 0.001).

Complication	Relevant to the Total Number of Patients: 290 (100%)	Relevant to the CP Group: 136 (100%)	Relevant to the ED Group: 154 (100%)	*p*-Value (ED vs. CP)
Number of complications (mean):	50%	51%	64%	<0.001 **
Pseudarthrosis:	4% (n = 13)	1% (n = 2)	7% (n = 11)	<0.001 **
Necrosis and infection:	2% (n = 7)	3% (n = 4)	2% (n = 3)	0.711
Ligamentous or muscular injury:	6% (n = 16)	5% (n = 7)	6% (n = 9)	0.803
CRPS:	1% (n = 4)	1% (n = 1)	2% (n = 3)	0.625
Osteoporosis by inactivity:	5% (n = 14)	4% (n = 6)	5% (n = 8)	0.79
Post-traumatic arthritis:	18% (n = 37)	7% (n = 10)	18% (n = 27)	0.013 *
Dead prior final examination:	0% (n = 0)	0% (n = 0)	0% (n = 0)	1.00
Secondary dislocation:	6% (n = 16)	5% (n = 7)	6% (n = 9)	0.803
Psychological disease:	0% (n = 0)	0% (n = 0)	0% (n = 0)	1.00
Neurological disease:	5% (n = 15)	1% (n = 1)	9% (n = 14)	<0.001
Premature plate removal:	3% (n = 10)	3% (n = 4)	4% (n = 6)	0.753
Belated bone healing:	3% (n = 10)	3% (n = 4)	4% (n = 6)	0.753

**Table 3 biomedicines-11-02862-t003:** Overview of the bone healing process post-surgery demonstrates some significant differences between the group of empty defect treatment (ED) and calcium-phosphate based augmentation. (* in case of significance with *p* < 0.05, ** in case of high significance level with *p* < 0.001).

	Days Post-Surgery [Min:Max; Mean ± SD]	Number of Patients (n)	*p*-Value Fracture Edge (ED vs. CP)	*p*-Value Fracture Gap (ED vs. CP)	*p*-Value Articular Surface (ED vs. CP)
Post-Surgery Examination	0:7; 1.84 ± 1.18	282	0.53	0.04 *	<0.001 **
First Follow-Up Examination	10:79; 27.06 ± 11.19	254	<0.001 **	<0.001 **	<0.001 **
Second Follow-Up Examination	40:131; 61.99 ± 20.83	233	<0.00 **	<0.001 **	<0.001 **
Third Follow-Up Examination	126:193; 106.84 ± 34.94	171	<0.001 **	<0.001 **	<0.001 **
Fourth Follow-Up Examination	126:571; 285,26 ± 76.40	126	<0.001 **	<0.001 **	0.002 *
Fifth Follow-Up Examination	234:1986; 645.02 ± 326.48	98	0.126	0.02 *	0.001 *

## Data Availability

In case of data requests, the authors will provide the data.

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
