# Peer review of "Traumatic Fracture Treatment: Calcium Phosphate Bone Substitute Case–Control Study in Humerus, Radius, Tibia Fractures—Assessing Efficacy and Recovery Outcomes"

_biomedicines, 2023, doi:10.3390/biomedicines11102862_

Round 1
Reviewer 1 Report
Suggestions for Improvement:
Clarity and Organization: The manuscript could benefit from improved organization and clarity. Consider rephrasing certain sentences to enhance readability. Additionally, clearly delineate different sections (e.g., Introduction, Methods, Results) to facilitate easier navigation for readers. Moreover, the description of the research requires substantial improvements regarding the type of the research- prospective, retrospective, sampling, and group allocation strategy. Provide specific details regarding comorbidities. Table S2 should be considered as a supplementary file. Please provide specific details regarding the clinical and paraclinical complications ( like posttraumatic arthrosis) assessments.
Statistical Analysis: Provide more detailed information about the statistical methods used for data analysis. Explain how Mann-Whitney tests, Kolmogorov analysis, and t-tests were applied, and justify the choice of significance level (p≤0.005) for some analyses.
Data Presentation: Use graphs and tables to visually represent complex data, such as the frequency analysis of severity and cause of injury. Visual aids can help readers better understand the results and comparisons. Replace Figures two and four with histograms. The results data should be better understood if the authors replace the text with tables which summarise and encompass the research results.
Except for imagistic results, no other assessments were performed.
Discussion: Extend the discussion of clinical implications and potential applications of the proposed classification system. Discuss the limitations of the study and possible avenues for future research.
Please consider adding limitations regarding:
Lack of Randomization: The study does not mention randomization or blinding procedures, which could introduce selection bias and affect the comparability of the treatment groups.
Single-Centre Study: The research is conducted at a single centre, which may limit the external validity of the findings. Multi-centre studies could provide more representative and generalizable results.
Lack of Long-Term Follow-Up: The study mentions a mean period of radiographs after 1.84 days post-surgery and a follow-up examination after almost two years. Longer-term follow-up data could provide insights into the durability and long-term outcomes of the treatment approaches.
Limited Clinical Parameters: The text focuses on radiographic and bone healing outcomes, but other important clinical parameters such as functional outcomes, patient-reported outcomes, and quality of life measures are not extensively discussed.
Also, I must underline the following aspects which requires improvements:
Incomplete Data Presentation: The excerpt provided is a partial text, and critical details such as patient demographics, comorbidities, specific fracture types, and other potential confounding factors are not fully presented. This makes it challenging to evaluate the study's methodology and results thoroughly.
Limited Comparison: While the study compares outcomes between the ED and CP groups, there is no direct comparison with other established treatments or bone graft substitutes, which could provide a more comprehensive understanding of the potential advantages or disadvantages of the interventions.
Author Response
The authors thank the reviewer for this critique work.
Sincerely

Reviewer 2 Report
This is an interesting study. I believe that calcium phosphate-based bone substitute materials is safe in clinical applications and it can improve early osseus healing process.
What was the authors' hypothesis?
What were patients' demographic data? Are there any differences between the study groups?
What was the impact of comorbidities?
Any multivariable analysis to control confunding factors?
A better discussion on the limitations of the study should be presented.
Author Response

(The authors gave the same response as above.)

Reviewer 3 Report
I feel that this articles requires lots of improvements:
Line 90-95- The text is repetition to what in Table S1. would be better if describing the comparison between calcibon and adult bone eg. Calcibon has smaller open porosity etc.
Line 90 (less resorption of the bone graft substitute was observed in Calcibon?)
Line 125- discuss why exclude bone tumour defect in discussion
Line 28- what are FU examinations?- State in full the first time you mention it, not in Line 322.
Line 170-178- readers are confused with the mentioned number eg. (5), (4), (3) etc. in the text.
Line 237-238- please explain further
Line 239- “The duration in the hospital was [0:72; 10.66±11.51]” which one is minimum: maximum? I advise you present all data currently in [Minimum: Maximum; Mean±SD] to “mean±SD (minimum: maximum)” form
Please justify the need to report minimum: maximum. I find it is unnecessary to present minimum: maximum together with mean±SD. People usually present mean with SD, median and IQR
Figure 2 describe lots about nature of the fracture. Is there any correlation drawn between nature of the fractures to any parameters? What is the importance of analysing nature of the fracture?
Figure 4- unnecessarily take up so much space. Figure 4 a-h can be presented in form of 1 or 2 bar graphs
Table S3- why number of complications is in decimal? Or is it actually proportion/ percentages?
Line 314-333- I think the main data in this article is data on bone healing. Why data on bone healing is not presented in Table/ figure?
Line 334- Why do you have another subgroup (aged >64 y.o.) to be analysed? What research question you are trying to address? Is it health outcome in elderly would be Calcibon? Answer this in the discussion section.
Require lots of revision on the language aspect especially in the Results section.
Choices of words are not good eg. Table S1 and Table S3
Line 243- 249- please revise the sentences. They are very confusing
Spelling mistake eg. Line 253
Line 66 (In contrast, in vitro studies have reported a high risk of infection [16] )- don’t understand, appears to be out of context
Line 69 (In vitro and animal studies exist for most materials.)- please elaborate further
Line 83-86 (The similarity of Calcibon® to natural bone is maximized using different minerals – alpha-tricalcium phosphate (alpha-TCP), calcium hydrogen phosphate (CaHPO4), calcium carbonate (CaCO3), and hydroxyapatite.) - do you mean highest bone biomimetic could be determined by having different combinations of materials? Please rephrase
Table S1- replace the word Parameters with Characteristics
Line 140- the word “similar gender” gives impression paired-wise comparison eg. paired t-test etc. please remove the word “similar” unless you intended to do paired-wise comparison.
The Discussion section was poorly written. The authors didn’t discuss the results much and relate the findings with literature. Very few references are cited in the Discussion section
Author Response

(The authors gave the same response as above.)

Round 2
Reviewer 1 Report
The authors performed substantial improvements of their manuscript and I consider it suitable for publication.
Author Response
Dear reviewer,
we thank you for your investigation and the trust in our work. Due to a reviewer´s opion, we enhanced some more facts in the manuscript.
Sincerely,
the authors.
Reviewer 2 Report
Major suggestions have been fulfilled.
Author Response

(The authors gave the same response as above.)

Reviewer 3 Report
While there has been many comments already addressed, there are still few yet to be addressed:
-discuss why exclude bone tumour defect in discussion —> not addressed
- “The duration in the hospital was [0:72; 10.66±11.51]” which one is minimum: maximum? I advise you present all data currently in [Minimum: Maximum; Mean±SD] to “mean±SD (minimum: maximum)” form—> not addressed
Please justify the need to report minimum: maximum. I find it is unnecessary to present minimum: maximum together with mean±SD. People usually present mean with SD, median and IQR —> not addressed
- Why do you have another subgroup (aged >64 y.o.) to be analysed? What research question you are trying to address? Is it health outcome in elderly would be Calcibon? Answer this in the discussion section.
—> not addressed
Require substantial revision on language eg. Line 55-56 “The advantages of synthetic bone substitutes include an unlimited availability and an availability at any time without necessitating a second surgical site [6].”
Obvious mistake on spelling eg. inflamm aging (Line 471)
Author Response
Dear reviewer,
we thank you for your investigation and the trust in our work. We enhanced the manuscript. Please look at the attached letter for the answers to your opinion.
Sincerely,
the authors.

Round 3
Reviewer 3 Report
Satisfied with the revision/ amendments made.